# Genetic Adaptation of Coxsackievirus B1 during Persistent Infection in Pancreatic Cells

**DOI:** 10.3390/microorganisms8111790

**Published:** 2020-11-15

**Authors:** Anni Honkimaa, Bryn Kimura, Amir-Babak Sioofy-Khojine, Jake Lin, Jutta Laiho, Sami Oikarinen, Heikki Hyöty

**Affiliations:** 1Faculty of Medicine and Health Technology, Tampere University, 33520 Tampere, Finland; bryn.kimura@tuni.fi (B.K.); amirbabak.sioofykhojine@tuni.fi (A.B.S.-K.); jutta.laiho@tuni.fi (J.L.); sami.oikarinen@tuni.fi (S.O.); heikki.hyoty@tuni.fi (H.H.); 2Finnish Institute of Molecular Medicine (FIMM), University of Helsinki, 00290 Helsinki, Finland; jake.lin@helsinki.fi; 3Fimlab Laboratories, Pirkanmaa Hospital District, 33520 Tampere, Finland

**Keywords:** type 1 diabetes, enterovirus, coxsackievirus B1, next generation sequencing, persistent infection, cell models of persistency, virus adaptation

## Abstract

Coxsackie B (CVB) viruses have been associated with type 1 diabetes. We have recently observed that CVB1 was linked to the initiation of the autoimmune process leading to type 1 diabetes in Finnish children. Viral persistency in the pancreas is currently considered as one possible mechanism. In the current study persistent infection was established in pancreatic ductal and beta cell lines (PANC-1 and 1.1B4) using four different CVB1 strains, including the prototype strain and three clinical isolates. We sequenced 5′ untranslated region (UTR) and regions coding for structural and non-structural proteins and the second single open reading frame (ORF) protein of all persisting CVB1 strains using next generation sequencing to identify mutations that are common for all of these strains. One mutation, K257R in VP1, was found from all persisting CVB1 strains. The mutations were mainly accumulated in viral structural proteins, especially at BC, DE, EF loops and C-terminus of viral capsid protein 1 (VP1), the puff region of VP2, the knob region of VP3 and infection-enhancing epitope of VP4. This showed that the capsid region of the viruses sustains various changes during persistency some of which could be hallmark(s) of persistency.

## 1. Introduction

Coxsackie B group viruses (CVBs) belong to Enterovirus (EV) genus in the family of *Picornaviridae*. EVs have positive-sense single-stranded RNA genomes, approximately 7400 nucleotides (nt) long, inside a 30 nm icosahedral protein capsid [1]. The protein capsid consists of 12 pentameric subunits, built up of five protomers. Each protomer contains viral structural proteins VP1-4. EVs have fivefold, threefold, and twofold symmetry axes. The VP1 protein is located around fivefold axes, whereas VP2 and 3 are found around threefold axes. EVs have a deep depression, called the canyon, which encircles the fivefold axes below VP1. Often the canyon is a receptor binding site [2,3]. A possible externalization site for the RNA genome is located near the twofold axes [4].

The EV genome was long thought to contain only a single open reading frame (ORF), which is proteolytically processed into four structural proteins (VP1-4) and into seven non-structural proteins (2A-2C and 3A-3D). However, recent studies have identified another ORF harbored upstream, hereafter called the second ORF (ORF2) [5,6]. The ORF2 is translated into a protein called ORF2 protein (ORF2p), which has been proposed to have a role in virus growth in gut epithelial cells. The viral genome has untranslated regions (UTRs) at each end. The 5′-UTR is long (740 nt), highly conserved and it has important functions in viral replication. It contains the internal ribosome entry site (IRES) of the virus and is covalently attached to a small protein called viral protein genome linked (VPg) [7]. The 3′-UTR is flanked by a poly-A tail and is important in (-) strand synthesis [8].

Typically, CVBs cause acute illnesses such as common cold, encephalitis, meningitis, myocarditis, herpangina, pancreatitis and hepatitis [9]. Additionally, CVBs have been shown to establish persistent infections [10,11,12,13] which have been linked to the pathogenesis of certain chronic diseases including e.g., chronic cardiomyopathies, type 1 diabetes (T1D) and post-polio syndrome [14,15,16,17].

The association between CVBs and T1D has been shown in several studies including serological studies and direct virus detection [18,19,20,21]. CVBs include six different serotypes (CVB 1-6) and CVB1 is one of the serotypes that has associated with increased risk of T1D [19]. Currently, the most plausible hypothesis is that a persistent infection in the insulin-producing pancreatic beta cells leads to cell-damage either by a direct viral effect or via virus-induced inflammation [22,23,24,25,26].

EVs have previously been thought to cause lytic infections. However, it has become evident that EVs can also persist in cells and tissues [10,13,23,27]. The exact mechanism of EV persistency in cultured cells is not known, but seemingly it requires co-evolution of both the host cell and the infecting virus [11,28]. Two different types of EV persistency have been proposed to exist, a carrier state and a steady state persistency. In carrier state persistency, only a small proportion of the cells are infected, but the virus is produced in relatively high titers [11,28]. In steady state persistency, on the other hand, nearly all the cells are infected, but the virus is unable to complete lytic replication [29]. Steady state persistency has been linked to terminal deletions, of 7 to 49 nts long, in the 5′-UTR of the viral genome [27].

We have previously characterized cellular adaptation during the development of CVB1 persistency using proteomics approach [30]. This study aims at identifying the changes that occur in the CVB1 genome during the development of persistence. To this end, we have established persistent infection in pancreatic ductal cell line (PANC-1) with four strains of CVB1, and in pancreatic beta cell line (1.1B4) with three strains of CVB1 to identify mutations that are common in all these persistence models. The genome modifications of each persistent infection-derived virus (PIDV) were characterized using next generation sequencing methods (NGS).

## 2. Materials and Methods

### 2.1. Persistent Infection

Four strains of CVB1 were used to establish altogether eight persistent models in two pancreatic cell lines: in pancreatic ductal cell line, PANC-1, and in pancreatic beta cell line, 1.1B4 (mycoplasma-free cell lines were obtained from American type culture collection, ATCC, Manassas, VA, USA). The four CVB1 strains included Conn-5 a prototype strain, later referred as ATCC (obtained from the ATCC) and three clinical isolates 10796 (Argentina 1983), 10797 (United States, Connecticut, 1984) and 10802 (Argentina 1998) [31]. All four strains were used to establish persistent infection in PANC-1 cells and three of the strains CVB1 ATCC, 10796 and 10802 were used to establish a persistent infection in 1.1B4 cells. CVB1 ATCC virus has established duplicated biological model on PANC-1 cells named E1 and E2. The PIVD are named as “strain code-PIDV_[Cell line]_” in this publication.

The persistent infection was established using a protocol earlier described by Sane et al. for CVB4 E2 [10] also as described in our previous publications [30,32]. After the initial infection the cells were washed three times a week. The cells were harvested once a week after they had started to grow after the initial lytic phase of infection. In addition, in the CVB1 ATCC and 10796 persistent infection models on PANC-1 cells, an additional transfer of cells on fresh PANC-1 cells was done 2–4 weeks after the initial infection. Presence of the virus genome and infectious virus particles were verified using RT-qPCR [33,34], and virus plaque formation [18] from different timepoints during persistency. The plaque morphology was also studied.

#### Immunofluorescent Labeling

Virus infected cells were identified by immunofluorescent labeling. Persistently infected cells were harvested 3.4–4.7 and 12.7–14.2 months post-infection by scraping, and cell pellets were fixed in 10% formalin (Oy FF-Chemicals Ab, Haukipudas, Finland) for 24 h prior to dehydration and paraffin embedding. The formalin-fixed paraffin-embedded (FFPE) samples were immunostained using two different antibodies; an in-house generated monoclonal rat antibody, clone 3A6, (kindly provided by professor Vesa Hytönen, Tampere University), which identifies an epitope in the N-terminal region of EV-VP1 [35], and J2, a commercial monoclonal mouse antibody (SCICONS English & Scientific Consulting Kft., Szirák, Hungary) [36], which detects virus dsRNA (a byproduct of the virus replication). Commercial goat anti-rat 568 and anti-mouse 488 IgG (H + L) Alexa Fluor (Thermo Fisher Scientific, Waltham, MA, USA) secondary antibodies were used for detection. Imaging was performed using an Olympus BX-60 microscope.

### 2.2. Next Generation Sequencing

Cell culture supernatants were collected from three time points: one day post infection (time point 0; TP0), and approximately 5–7 and 12-months post infection TP1 and TP2, respectively. Virus RNA was extracted using QIAamp Viral RNA Mini kit (Qiagen, Hilden, Germany) according to the manufacturer’s instruction directly from cell culture supernatant. Sample preparation and NGS were performed as described elsewhere [37,38]. The RNA was reverse transcribed with ImProm II reverse transcriptase (Promega, Madison, WI, USA), the second strand synthesis was performed by adding 5’-end exonuclease negative Klenow fragment (Promega, Madison, WI, USA). After purifying unincorporated primers and single stranded molecules the double-stranded transcripts were amplified using their respective invariant tags as primers by a 20 cycles’ PCR with the AmpliTaq Gold chemistry (Applied Biosystems, Foster City, CA, USA) [37]. Size selection and purification of the pre-amplified libraries were done using Agencourt AMPure XP (Beckman Coulter, Fullerton, CA, USA) mixed with the product at 0.5× ratio. Nextera XT DNA Sample Preparation Kit (Illumina, San Diego, CA, USA) was used to process the libraries in batches of twenty-four samples per sequencing run. The protocol provided by the manufacturer was followed until the normalization step. The libraries were diluted for sequencing according to concentrations measured by KAPA kit (Illumina, San Diego, CA, USA) before and after the pooling, and were denaturized prior to sequencing. Sequencing of the libraries was done on a MiSeq instrument (Illumina, San Diego, CA, USA) using the Nextera XT v2. 250 × 250-cycles sequencing kit (Illumina, San Diego, CA, USA).

### 2.3. Analyzing the Sequence Data

The quality of the sequencing data was determined using FastQC-0.11.7 [39] and the sequencing reads were trimmed according to the quality using Trimmomatic-0.36.0 [40]. BBTools-37.96 [41] Clumpify tool was used to remove PCR and optical duplicate reads. The reads were then aligned with BWA-0.7.17 [42] to strain-specific reference sequences which were obtained through Sanger sequencing on the parental virus strains. The alignments were filtered for mapping quality and sorted using Samtools-1.6. [43] while Breseq-0.31.1 [44] was run in polymorphism mode for single nucleotide polymorphism (SNP) discovery. A threshold of 5 forward and 5 reverse-strand reads containing the alternate allele was used before a SNP was called. In addition, the alternate allele had to appear in at least 5% of the reads covering the position. Synonymous SNPs were filtered out with custom scripts. Lolliplots visualizing SNPs across the genome were created with R package Trackviewer-1.16.0 [45]. Nucleotide sequences were aligned using Clustal omega online tool [46]. Nucleotide sequences were translated into amino acid sequences and were analyzed using GeneDoc program [47].

### 2.4. 3D Modelling

The 3D structure was modelled based on the Sanger sequencing of the CVB1 ATCC parental virus. At the time of initial analysis, a CVB1 3D template was not available, however, CVB3 is structurally similar to CVB1 therefore two CVB3 3D templated were used for modelling [48]. The capsid was 3D modelled using CVB3 X-ray diffraction templates 4GB3 [49] and 1COV [48] obtained from Protein Data Bank (PDB) [50]. Modelling software Modeller-9.20 was used to create 20 models [51]. The model with the lowest discrete optimized protein energy (DOPE) score, representing the highest structural quality, was chosen for visualization [52]. The model is visualized in ChimeraX.

## 3. Results

### 3.1. Characteristics of Established Persistent Infections

Altogether eight persistent infection models were established in two pancreatic cell lines (PANC-1 and 1.1B4), using four CVB1 strains: ATCC prototype strain (Conn-5), and three clinical isolates called 10796, 10797 and 10802. The genome copy number generally decreased immediately after the initial infection due to cell death, but gradually increased to a steady level of 10^8^–10^9^ copies/mL in the supernatant (Figure 1).

Virus protein and dsRNA were detected in proportion of the cells by immunostaining in time points 1 and 2, 3.4–4.7 and 12.7–14.2 months post-infection respectively (Figure 2a). The amount of VP1 positive cells in persistent infection has shown to fluctuate over time [53]. Virus positive cells were detected in all virus infected cell cultures in both time points despite the mutations that occurred in viral genome at the suspected 3A6 antibody binding site [35]. One mutation was identified in this site in ATCC E2-PIDV_[PANC-1]_, 10796-PIDV_[PANC-1]_, 10796-PIDV_[1.1B4]_, and 10802-PIDV_[1.1B4]_ strains, and three mutations in ATCC-PIDV_[1.1B4]_ strain (Figure 2b). However, the effect of these mutations on the antibody binding was not quantified. In addition, the amount of infective virus from the culture medium of CVB1 ATCC E1, E2 and 10796 persistent infection models (PANC-1 cells) was analyzed using TCID_50_ assay showing the presence of infective virus in high titer (Appendix A).

#### Virus Plaque Morphology

Morphology of the virus plaques was identified in green monkey kidney cells (GMK) during the development of persistent infection. The plaque morphology naturally varies between CVB1 stains, which is seen in the first row in Figure 3. Generally, the size of the plaques decreased during the development of viral persistency in all PIDVs, except in ATCC-PIDV_[1.1B4]_ which didn’t show any obvious change even one-year post infection (Figure 3). The change in plaque morphology was more prominent in clinical isolates (10796, 10797 and 10802), which originally produced larger plaques than the ATCC strain (Figure 3).

### 3.2. Determining Genomic Sequences by NGS

The average read depth ranged from 214 to 12,308 with a median average of 1934. The percent of genome covered by the 10 read threshold for calling mutations ranged from 97–99% with a median percent coverage of 98%. No substitutions, insertions, or deletions were observed. The proportion of CVB1/human reads was 13.01% in average in the supernatant samples collected (Appendix A).

#### 3.2.1. Untranslated Regions

The read depth in the 3′-UTR was mainly below 10 in PIDV genomes (Figure 4) obtained from NGS analysis making variant calling not possible. The read depth was above 10 in the majority of the 5′-UTR, except in the beginning in domain I, also called as cloverleaf region (Figure 4). Domains II-V covers the internal ribosome entry site (IRES) that controls translation [7]. The mutations observed are randomly distributed, but majority of the mutations locates at positions 602–655, and at positions 691–717 (Figure 5).

#### 3.2.2. Accumulation of Amino Acid Substitutions in Structural Proteins

Amino acid (AA) sequence of each PIDV from TP2 was aligned to the parental virus strain and CVB3-Nancy (obtained from NCBI GeneBank, JX312064.1) for comparison. CVB3 was included in the analysis for its sequence similarity and known features from the literature. Non-synonymous AA substitutions were identified, in eight PIDV strains, from each viral structural protein compared to the relevant parental strains. The majority of the observed substitutions did not map to the same residues in all eight PIDVs, but the mutations accumulated in the similar regions of viral structural proteins.

The most interesting substitution was the K257R, which was the hallmark of persistency in all PIDVs at TP2. This AA is located towards the C-terminus of the VP1 the alternate AA before the K259 residue which is marked as a Coxsackie and Adenovirus receptor (CAR) binding AA footprint, a location which is a published immunogenic peptide (PEP91) in CVA-16 [55]. This might indicate an important relevant mechanism for persistency, but we did not confirm it by experiments such as side-directed mutagenesis and persistency establishment in cell models.

In the VP1 structural protein the mutations accumulated to the N-terminal region of VP1 (the suspected 3A6 antibody binding site [35]) and in four additional positions, including the region surrounding the BC loop (residues 81–85), the DE loop (residues 125–137), the EF loop (residues 145–165) and the C-terminus of VP1 (Figure 6). All PIDVs had at least one mutation in all of these three “hot spots”, except 10796-PIDV_[1.1B4]_ which did not sustain any non-synonymous mutations around the BC loop region, and 10802-PIDV_[PANC-1]_ which did not have any non-synonymous mutations in the EF loop region. Additionally, mutation R95K was detected in three PIDVs (ATCC E2-PIDV_[PANC-1]_, 10796-PIDV_[PANC-1]_ and 10797-PIDV_[PANC-1]_) and V97A in two of the PIDVs (ATCC E1-PIDV_[PANC-1]_ and ATCC-PIDV_[1.1B4]_) in the CD loop region (residues 91–97). ATCC E2-PIDV_[PANC-1]_ had a N202K mutation, 10802-PIDV_[PANC-1]_ mutations R201K and N202K, and 10796-PIDV_[1.1B4]_ had mutation N202K, which located in the GH loop region (residues 192–214). ATCC E1-PIDV_[PANC-1]_ had E223D and G227S mutations and 10796-PIDV_[PANC-1]_ A224V mutation in HI loop region (residues 222–228) respectively. Interestingly all the strains at TP0 had similar AA sequences in the BC, DE, and EF loops, while the persistent viruses had changed in this region.

Shown in Figure 7a, AA substitutions accumulated in the EF loop region of the VP2 capsid protein, also known as the “puff region” (residues 129–180). All CVB1 PIDVs had at least one mutation in the puff region and the majority of the mutations located to residues 158-167 within this region. Mutation G236R was found in two PIDVs (10797-PIDV_[PANC-1]_ and 10802-PIDV_[1.1B4]_) and P239S in two PIDVs (10796-PIDV_[PANC-1]_ and ATCC-PIDV_[1.1B4]_) in HI loop region (residues 233–239 in VP2). Additionally, five PIDVs had a mutation in AA70 in Βb-sheet (Q70H in 10796-PIDV_[PANC-1]_, 10797-PIDV_[PANC-1]_, 10802-PIDV_[PANC-1]_ and 10796-PIDV_[1.1B4]_; Q70Y in ATCC-PIDV_[1.1B4]_).

The PIDVs, except 10802-PIDV_[PANC-1]_ and 10802-PIDV_[1.1B4]_, had mutations which accumulated around the major protrusion on the VP3 capsid protein (knob region, residues 58–66 in VP3) and especially on AA58 in VP3 (Figure 7b). Additionally, the mutation in AA93 occurred in five of the PIDVs (G93S in 10796-PIDV_[PANC-1]_ and 10796-PIDV_[1.1B4]_; N93S in 10797-PIDV_[PANC-1]_, 10802-PIDV_[PANC-1]_ and 10802-PIDV_[1.1B4]_) in the CD loop region (residues 86–97 in VP3). Also, V141A mutation was in three PIDVs including ATCC E1-PIDV_[PANC-1]_, 10796-PIDV_[PANC-1]_ and 10802-PIDV_[PANC-1]_, locating in the EF loop region (residues 136–144 in VP3). L117F mutation in Βd-sheet occurred in ATCC E1-PIDV_[PANC-1]_, ATCC-PIDV_[1.1B4]_ and 10802-PIDV_[1.1B4]_.

Again, the AA sequences of the puff and knob regions in the parental strains were identical and showed changes upon persistency establishment. Both CAR and a decay accelerating factor (DAF) binding extensively map to these two regions and the changes in these regions might have played a role in establishing or as the result of persistency.

Five out of eight PIDVs had mutations in the infection-enhancing epitope (AA11-30) [56,57] on VP4 capsid protein (Figure 7c) (stains ATCC E2-PIDV_[PANC-1]_, 10796-PIDV_[PANC-1]_, 10797-PIDV_[PANC-1]_, 10802-PIDV_[PANC-1]_ and 10796-PIDV_[1.1B4]_). AA16-24 have been reported to be the most variable sequence in this region, which was also seen in these PIDVs (Figure 7c) [58].

#### 3.2.3. Accumulation of Amino Acid Substitutions in Non-Structural Viral Proteins

The non-structural proteins including 2A-C and 3A-D had less mutations than the structural proteins. Only a few of the mutations in 2A-C were shared among different PIDVs (Appendix A). In 2A two PIDVs (ATCC E1-PIDV_[PANC-1]_ and ATCC-PIDV_[1.1B4]_) had a mutation in AA11 (respectively Y11C and Y11H) and one PIDV (10802-PIDV_[PANC-1]_) had a close mutation, N14T. The 10796-PIDV_[PANC-1]_ had six mutations in 2A, whereas other PIDVs had only 1–3 mutations. The mutations of 10796-PIDV_[PANC-1]_ included a small cluster of mutations in AA86-94 (E86G, Y89H and Y94H).

In 2B there are very few mutations (Appendix A). One mutation, S15L, was observed in all three CVB1 ATCC PIDVs and one of the mutations was located at the hydrophobic domain, but the mutation was from a hydrophobic AA to another hydrophobic AA (I50V). One mutation was located in the hydrophilic domain (K58R, changed to functionally similar AA). Additionally, three PIDVs have mutations in AA91-92.

In 2C protein there are only a few mutations, which include one mutation, M175L, at the ATPase motif B in 10796-PIDV_[PANC-1]_ and _[1.1B4]_ (Appendix A). Additionally, three PIDVs have mutations in AA91-92 and in AA251-259. None of these mutations were observed in the cis-acting replication element (CRE) in the 2C protein-coding region [CRE(2C)], in which the mutations have been shown to drive the de novo generation of genomic deletions at the 5′ terminus [59].

Among 3A-D, two of the proteins were conserved. The 3B protein (VPg), was conserved in all PIVDs. Also, 3C protein was conserved, since only one AA substitution (Q52R) was observed in the 10802-PIDV_[PANC-1]_. In the 3A protein only a few mutations were observed. However, one mutation (V58I) was in four PIDVs, including 10796-PIDV_[PANC-1]_, 10802-PIDV_[PANC-1]_, ATCC-PIDV_[1.1B4]_ and 10802-PIDV_[1.1B4]_. In addition, ATCC E1-PIDV_[PANC-1]_ carried the H57Y mutation, which has been linked to resistance against antiviral drugs including enviroxime, GW5074 and PIK93 [60]. Only a few mutations were observed in the 3D polymerase, but they were not shared by other strains and were randomly distributed in the genome. One mutation (V230I) was located in motif A in ATCC E1-PIDV_[PANC-1]_ and two mutations in motifs B: S299T in 10976-PIDV_[1.1B4]_ and I306V in ATCC E2-PIDV_[PANC-1]_.

#### 3.2.4. The Second ORF

The recently published second ORF was identified from CVB1 sequences. The second ORF is translated into the ORF2p protein. The protein has a highly conserved WIGHPV domain, which is important for ORF2p-dependent viral intestinal infection [5,6]. WIGHPV domain was identified in all four CVB1 strains but in the last timepoint one mutation (V18A) was observed in 10802-PIDV_[1.1B4]_ (Figure 8).

#### 3.2.5. Virus Surface Structure

The surface of CVB1 pentameric structure was 3D modelled with ChimeraX using the structure of CVB3 as a template (Figure 9).

CVB1, as well as all other CVBs, CAR for cell entry [61]. Besides CAR, some CVBs, like CVB1, interact DAF (also known as CD-55) [62]. CAR binding site is located in the canyon of the CVBs whereas DAF binding site is outside the canyon. The AAs interacting with these molecules are shown in Figure 9b [61,63,64,65]. The majority of the mutations in these footprint areas locate in DAF binding site (Figure 9c). In the CAR binding site, AA202 in VP1, was mutated (N202K) in three PIDVs (ATCC E2-PIDV_[PANC-1]_, 10802-PIDV_[PANC-1]_ and 10796-PIDV_[1.1B4]_) as marked by an arrow and the number 1 in Figure 9c. In the DAF binding site, AA162 in VP2, was mutated in five PIDVs (T162P in ATCC E1-PIDV_[PANC-1]_, ATCC-PIDV_[1.1B4]_; T162A in ATCC E2-PIDV_[PANC-1]_; E162G in 10796-PIDV_[1.1B4]_; E162Q in 10802-PIDV_[1.1B4]_), and AA237 in VP3 was mutated (Y237F) in three PIDVs (ATCC E1-PIDV_[PANC-1]_, ATCC E2-PIDV_[PANC-1]_ and ATCC-PIDV_[1.1B4]_) (Figure 9c). The VP2 mutation in AA 165 was shared by both these binding sites and found in three PIDVs (N165Y in ATCC E1-PIDV_[PANC-1]_ and ATCC-PIDV_[1.1B4]_; S165R in 10796-PIDV_[PANC-1]_) as indicated by an arrow (3.) in Figure 9c. Mutation K257R in VP1, found in all eight PIDVs (mentioned above), is located close to the receptor binding sites (Figure 9c). Additionally, the mutations in PIDV accumulated within the puff region are also the ones that interact with the CVB1 receptors.

## 4. Discussion

The results of the present study indicate that the CVB1 strains that have adapted to viral persistence differ from the parental CVB1 strains which have not been exposed to such selection pressure—they produced smaller plaques and shared common mutations in their genome. Since viral persistence was established in two different cell lines using four different CVB1 strains (altogether 8 models) we were able to study whether any mutation occur similarly in several persisting CVB1 strains. The mutation frequency on RNA viruses is relatively high, and ∼4 to ∼8 mutations/10^4^ nt is well-tolerated and are fully stable in the context of infectious virus, creating a concept called quasispecies. The upper limit of mutation frequency that the virus can tolerate without loss of infectivity was proposed to be in the range of 11–12 mutations/10^4^ nt [66]. Thus, there are numerous possibilities for virus changes during persistency. The virus may become less cytopathic e.g., by having impaired initiation of translation, adsorption (receptor binding), internalization (uncoating) or failure in efficient host shut off. Several studies have shown that mutations in structural proteins have an effect on virulence and pathogenicity. Even one amino acid change in structural proteins may modulate important functions in virus infection [67,68,69,70].

In the current study one mutation was particularly prominent—the amino acid substitution K257R in the VP1 protein was found in all these strains. This amino acid substitution has previously been reported in a non-lytic CVB6 Schmitt strain during a blind-passage in HPDE-cells when it acquired a lytic phenotype with increased production of progeny virus and increased number of infected cells [71]. This amino acid is located in the canyon region of the viral capsid. The authors found also some evidence suggesting that the mutation influenced viral interaction with DAF [71]. CVBs use CAR as the main receptor for internalization [61,72] but CVB1, CVB3 and CVB5 bind also to DAF [73,74]. CAR binding causes a conformational change in the viral capsid, called A-particle formation, which leads to externalization of the viral genome into the cytoplasm. However, DAF binding does not induce A-particle formation or cytolytic infection [75,76,77], but can trap the virus into the intercellular junctions where the CAR molecules locate [72]. However, this mutation is not directly located in the DAF-binding region of the viral capsid. On the other hand, some other mutations located in the DAF footprint area, including the mutation T162 in VP2, which occurred in five PIDVs. Mutations in the DAF binding-site have previously been shown to regulate the pathogenicity of CVB1 infection in a mouse model [78,79]. Three amino acids showed variation also in the CAR footprint area. One of these mutations (AA162 in VP2) occurred in five PIDVs. However, we have previously found that at least some of the PIDVs (ATCC E2-PIDV_[PANC-1]_ and 10796-PIDV_[PANC-1]_) are still able to use CAR as a receptor for internalization [30]. It is possible that some of the mutations that link to development of persistent CVB infection may regulate virus binding to the DAF and CAR molecules. Also, we do not know if both CAR and DAF binding are modified in persistency and virus can mutate back to parental strain upon acute infection.

In addition to the dominant K257R substitution in VP1, other mutations accumulated in structural proteins, including N-terminal region, a suspected 3A6 antibody binding site, BC, DE, EF loops and the C-terminus of VP1, the VP2 puff region, VP3 knob and the infection-enhancing epitope of the VP4 [56,57]. All these regions have been linked to virus biology in previous studies. The EF loop of VP2 (puff region; residues 129–180) and the surface protrusion of VP3 (knob region; residues 58–66) include the neutralizing immunogenic sites in rhinoviruses [80,81] and polioviruses [82,83]; and have receptor binding significance. Mutations in these regions have been shown to change the ability of CVBs to induce myocarditis [79,84,85] and can alter the virulence in mice. They also regulate plaque morphology in cell culture [86]. Mutation in these regions were common among studied PIDVs, since all of the PIDVs had mutations in the puff region and six of the PIDVs have mutations in the knob region. The VP4 substitution in the infection-enhancing epitope region in AA16 (S16R) has been linked to the plaque phenotype of CVB4 and to avirulence in vitro and intermediate phenotype in vivo (mice) [87,88]. Five of the PIDVs had mutations in infection-enhancing epitope, and particularly mutation at AA16, G16R, was observed in ATCC E2-PIDV_[PANC-1]_. Alidjinou et al. have studied carrier-state persistent CVB4 infection in PANC-1 cell model. They identified non-synonymous mutations in VP1 and VP2, structural proteins, and in non-structural proteins 2A, 2C and 3D [28], including altogether 105 mutations, of which 40 were amino acid substitutions occurring mainly in VP1 and 2A. No mutations were found in the CAR footprint region [28]. In the present study, the VP1 region has several mutations, and four residues with mutations accumulated in different PIDVs’. All the PIDVs had mutations in DE loop and in the N-terminus of the VP1. The DE loop is the most prominent exterior loop in VP1 (residues 125–137, in CVB3) found at the fivefold axis [48]. The DE loop may stabilize the capsid and is responsible for pH stability. This is because each DE loop interacts with other DE loops due to icosahedral symmetry; and because DE loops are involved in interactions with putative ion-binding sites along the fivefold axis. Seven of the PIDVs had mutations in the EF loop and surrounding the BC loop. The BC loop (residues 81–85 in CVB3), which flanks the rim of the canyon closest to the fivefold axes [48], has been shown to be an important neutralizing immunogenic sites for coxsackieviruses (CVB4) [89], polioviruses [82,83] and rhinoviruses [80,81]. Additionally five of the PIDVs had mutations mapped into N-terminal region which is suspected to be the 3A6 antibody binding site, and also it is known to be the binding site of the widely used EV antibody (Dako, clone 5-D8/1, Agilent) which could influence immunostaining of persisting EV in tissue samples. For example, 1.1B4 cells persistently infected with different CVB1 strains had less VP1 positivity in the last time point, which may be due to mutation in AA28.

Mutations were rare in non-structural proteins. Two of them, 3B and 3C, were conserved and remained intact. Additionally, the 3A had few mutations in some of the PIDVs. One of the 3A mutations, H57Y in ATCC E1-PIDV_[PANC-1]_, has been reported as linked to resistance to antiviral drugs [60]. The 2A-C had mutations among the studied PIDVs, however, no variable region was mapped in any of these proteins.

The 5′ UTR has domains I-VI and domain I is also called cloverleaf region, which plays a role in viral RNA replication [90,91]. In the current analyses the cloverleaf region the read depth was below 10 in PIDV genomes obtained from NGS analysis making variant calling not possible. The domains II-V have the internal ribosome entry site (IRES), having a role in translation [7]. The IRES has a polypyrimidine tract, where the 40S ribosomal subunits bind [92]. Mutations at IRES may influence virulence. As an example, in a site-directed mutagenesis study with CVB1, nt 573 and 579 within the 5′-UTR was mutated leading to significantly elevated virulence and pathogenesis of the virus [93]. Viral replication efficiency and viral virulence are also affected by proper folding and functioning of the 5′-UTR [54]. In the current study several mutations were mapped into 5′-UTR and majority of the mutations were not on highly reactive base pairs; however, the 10802-PIDV_[PANC-1]_ and _[1.1B4]_ both had mutation mapped on highly reactive base pair (nt 234) [54].

The second open reading frame has been recently identified in human enteroviruses (*Enteroviruses A*, *B* and *C*) [5,6]. It overturned the impression that EVs have only a single open reading frame. The second ORF is translated into second ORF protein (ORF2p), which is a membrane-associated protein that is localized to cytosolic vesicles. According to the latest studies ORF2p facilitates intestinal infection and release of the viral particles from intestinal epithelial cells [6]. The conserved WIGHPV domain, which is important for ORF2p-dependent intestinal infection [6], had a single AA substitution on the WIGHPV conserved domain V18A in one of the PIDVs (10802-PIDV_[1.1B4]_). However, overall, the ORF2p did not include any common mutations in persisting virus strains, but interestingly the ORF2p has many differences between CVB1 strains.

The study utilizes two pancreatic cell lines, 1.1B4 beta cell line and PANC-1 pancreatic ductal cell line, and four different CVB1 strains with different immunogenic properties [31]. We have previously observed that CVB1 10796 strain is weakly stimulating in plasmacytoid dendritic cells whereas CVB1 10802 strain is strongly immunogenic [31]. CVB1 10796 replicated also faster and produced significantly higher amount of virus than CVB1 10802. The ATCC strain induces modest and CVB1 10797 only weak innate immune response [31]. Altogether this study design, using different virus strains with different phenotypic characteristics and two cell lines, increased the probability that the mutations that developed in the majority of these models were linked to the development of persistency. These mutations are potentially attractive targets for future studies using site-directed mutagenesis and other tools to study their effect on virus biology in detail.

## Figures and Tables

**Figure 1 microorganisms-08-01790-f001:**
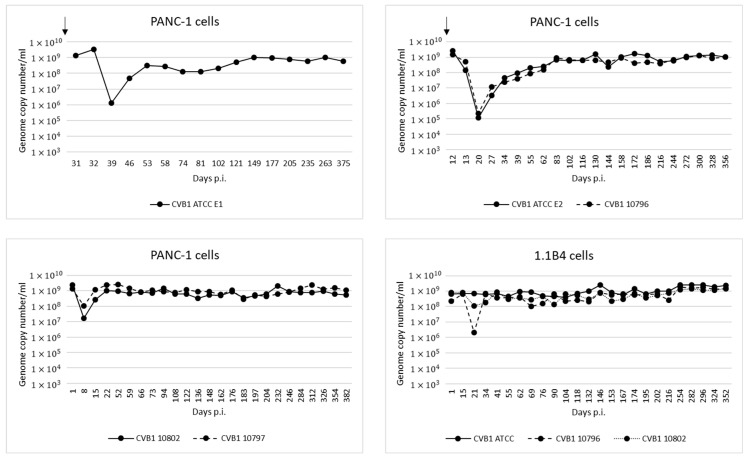
Genome copy numbers measured by RT-qPCR from cell culture supernatants collected from persistent infection models over time. In all the models the copy number eventually stabilized to approximately 10^8^ to 10^9^ copies/mL. Addition of fresh cells into three of the models is indicated by an arrow.

**Figure 2 microorganisms-08-01790-f002:**
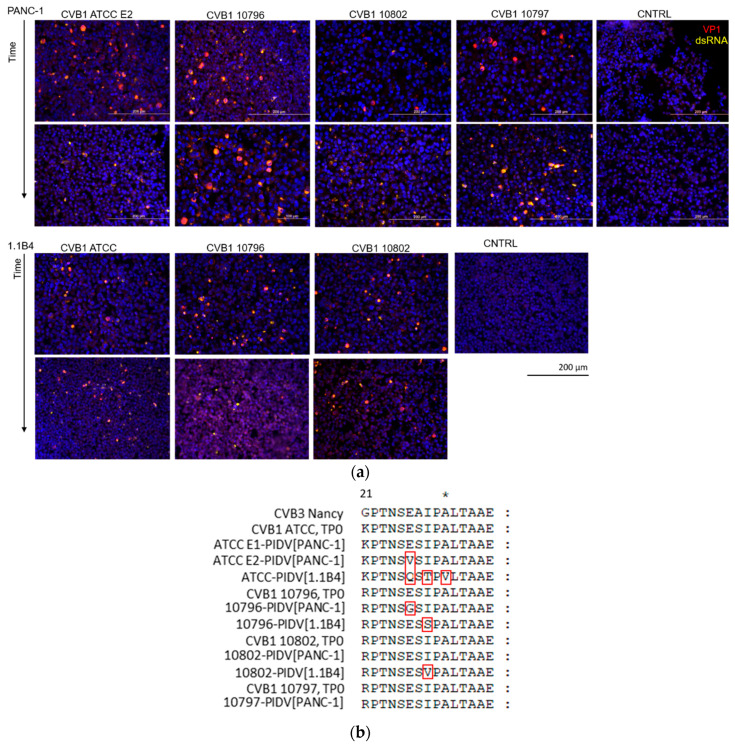
(**a**) Immunostaining of persistently CVB1 infected PANC-1 and 1.1B4 cells from two-timepoints (3.4–4.7 months post-infection in top row and 12.7–14.2 months post-infection in bottom row, corresponding to TP1 and TP2 respectively) and of uninfected control cells (CNTRL). VP1 protein (red) and dsRNA (yellow), and the average proportion of cells positive for viral VP1 protein was 22.9 ± 13.7% at the later time point (12.7–14.2 months post-infection) ranging from 7.1 ± 0.5% in the CVB1 10796 persistent infection model in 1.1B4 cells to 41.7 ± 9.8% in CVB1 10797 model in PANC-1 cells (altogether three microscopic fields were counted in each model including a total of 16,338 cells). (**b**) The virus amino acid sequence of the epitope region of the 3A6 antibody used in VP1 staining is shown for each PIDV. The red frames mark the mutations observed in CVB strains at last time point.

**Figure 3 microorganisms-08-01790-f003:**
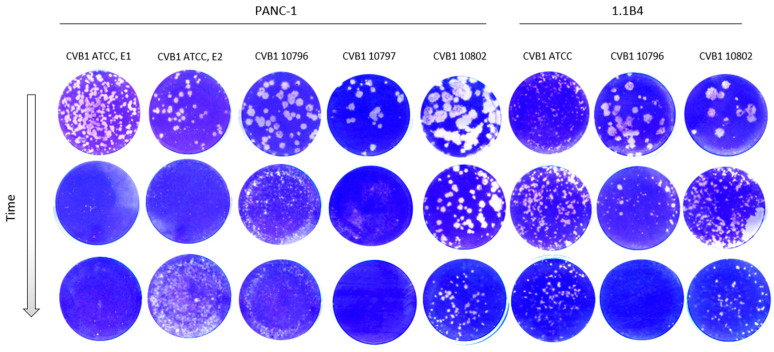
Plaque morphology of each PIDV over time. Plaque size reduces over time during the development of persistence in PANC-1 and 1.1B4 cell lines with the exception of ATCC-PIDV_[1.1B4]_ (6th column).

**Figure 4 microorganisms-08-01790-f004:**
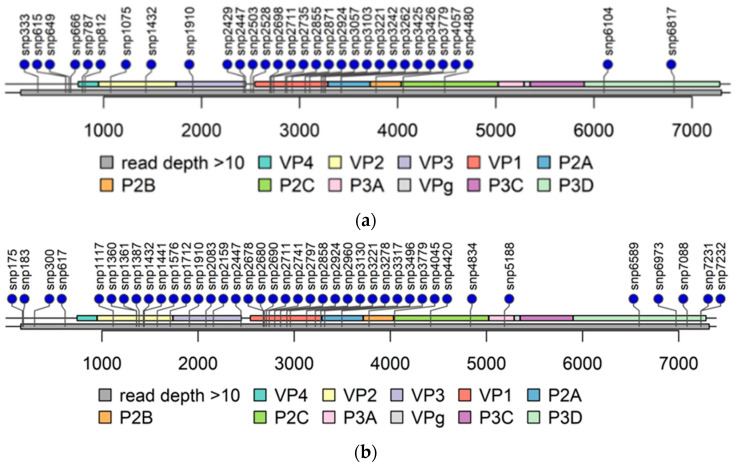
Lolliplots of each PIDV at the last timepoint. Single nucleotide polymorphism (SNP) are indicated by lollipop. All non-synonymous SNPs, and SNPs at both UTR are indicated. Grey bar below the genome bar characterizes the read depth (>10). (**a**) ATCC E1-PIDV_[PANC-1]_. (**b**) ATCC E2-PIDV_[PANC-1]_. (**c**) 10796-PIDV_[PANC-1]_. (**d**) 10797-PIDV_[PANC-1]_. (**e**) 10802-PIDV_[PANC-1]_. (**f**) ATCC-PIDV_[1.1B4]_. (**g**) 10796-PIDV_[1.1B4]_. (**h**) 10802-PIDV_[1.1B4]_.

**Figure 5 microorganisms-08-01790-f005:**
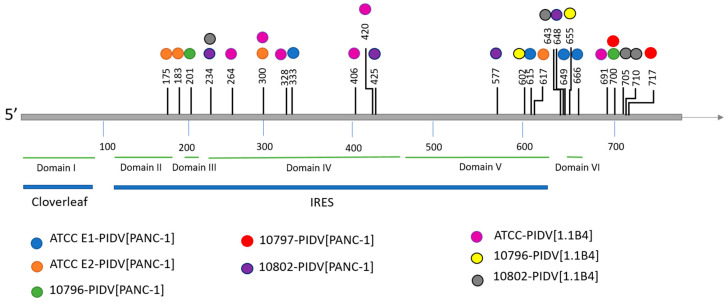
Mutation in 5′-UTR. SNPs are color coded indicating different PIDVs. Six distinct domains of the 5′-UTR are indicated by horizontal bars. Domain I is the cloverleaf structure and the IRES covers the domains II-V. Domain I: base pairs 2–87; Domain II: base pairs 105–181; Domain III: base pairs 189–209; Domain IV: base pairs 213–456; Domain V: base pairs 466–636 and Domain VI: base pairs 651–669 [54].

**Figure 6 microorganisms-08-01790-f006:**
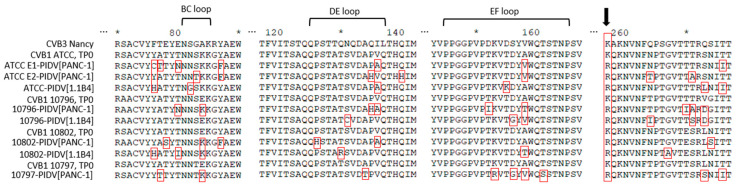
Non-synonymous mutations in PIDVs, marked with red frames, accumulated in the BC loop, DE loop and EF loop regions and at the end of the VP1 protein at TP2. Mutation K257R is indicated with an arrow.

**Figure 7 microorganisms-08-01790-f007:**
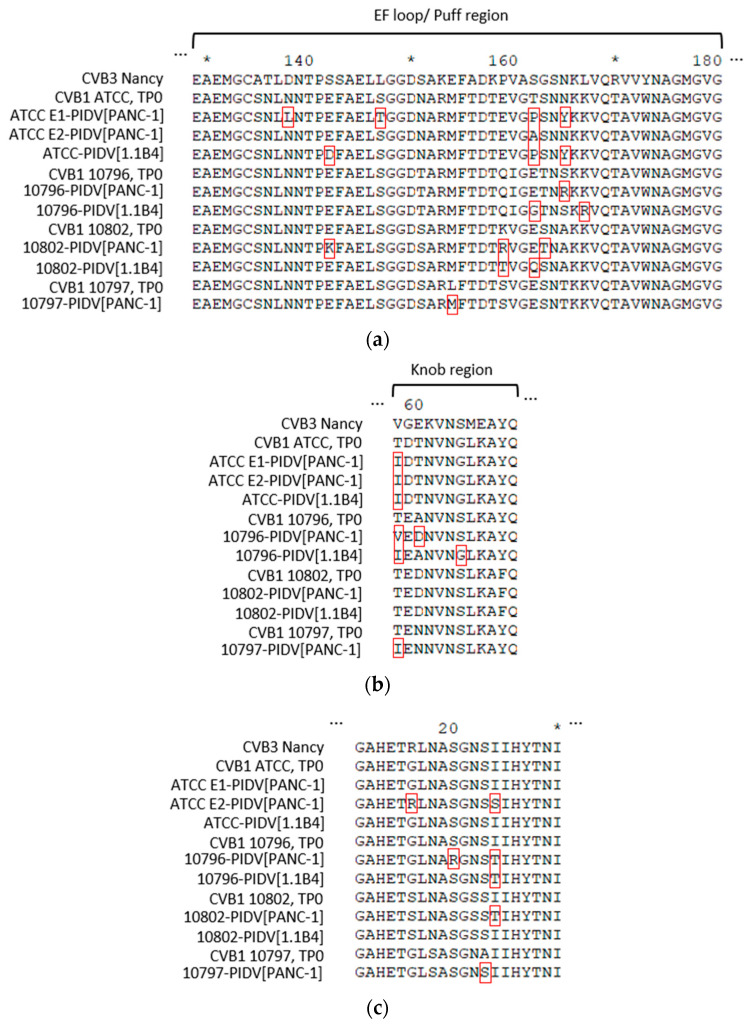
Mutations of PIDVs in capsid proteins VP2-4 in TP2. (**a**) Mutations on VP2 puff region. (**b**) Mutations on VP3 knob region. (**c**) Mutations on VP4 infection-enhancing epitope AA 11-30.

**Figure 8 microorganisms-08-01790-f008:**
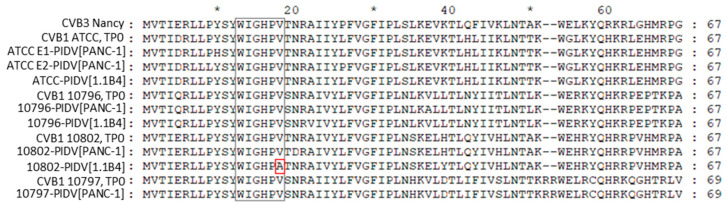
ORF2p and its conserved WIGHPV domain in TP2. One mutation was observed in conserved WIGHPV domain in 10802-PIDV_[1.1B4]_.

**Figure 9 microorganisms-08-01790-f009:**
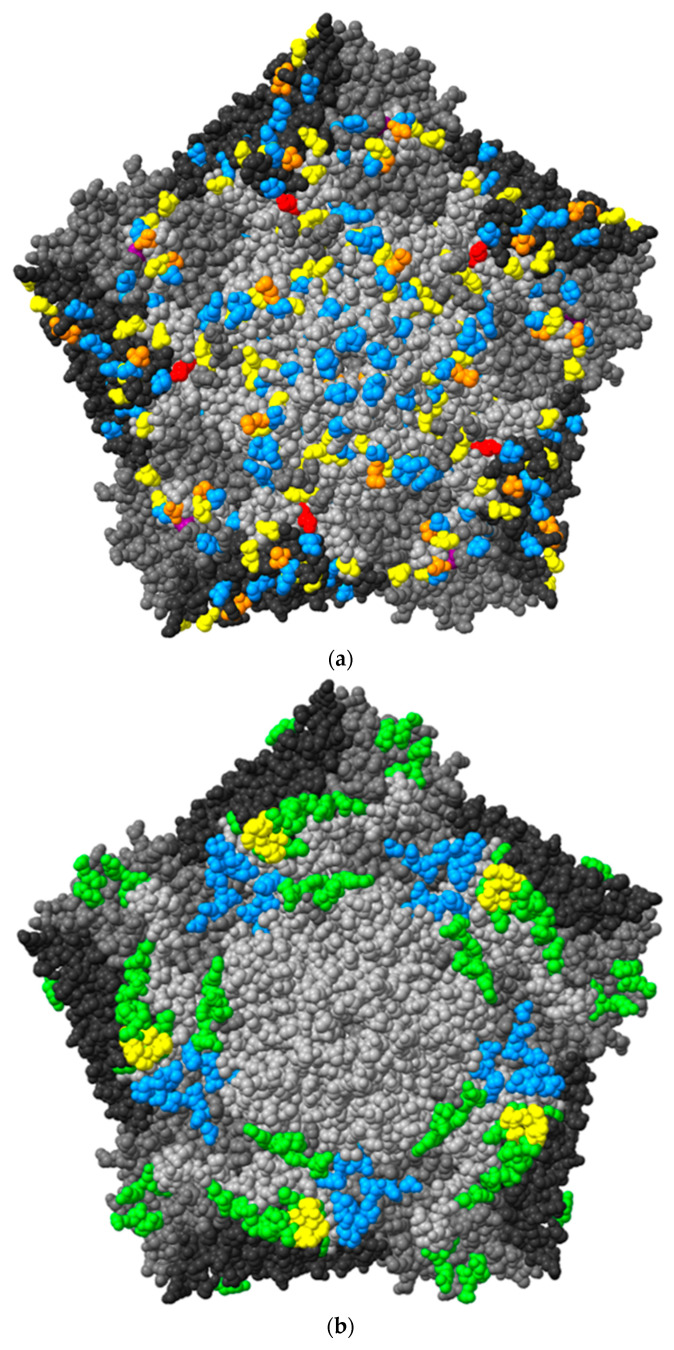
(**a**) 3D model of CVB1 virus pentameric structure displayed in ChimeraX from TP2. One pentamer consists of five protomers, each constructed by viral proteins VP1-4. Blue indicates mutation in one strain; yellow, mutations in 2–3 strains; orange, mutations in 4–5 strains; purple, mutations in 6 strains; and red, mutations (K257R) in the AA in all PIDVs. (**b**) 3D model of CVB1 with CAR binding site marked with blue color and DAF binding site with green color, shared AAs are marked with yellow color. (**c**) Mutations in CAR and DAF binding site are marked with red. Number 1. is mutation on AA202 in VP1 shared by three PIDVs, number 2. is mutation on AA 162 in VP2 shared by five PIDVs, number 3. is mutation on AA 165 in VP2 shared by three PIDVs and number 4. is AA 237 in VP3 shared by three PIDVs. AA257 in VP1, which has K257R mutation found in all studied PIDV strains, is also indicated.

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
