# Peer review of "Genetic Adaptation of Coxsackievirus B1 during Persistent Infection in Pancreatic Cells"

_microorganisms, 2020, doi:10.3390/microorganisms8111790_

Round 1

Reviewer 1 Report

Manuscript review: microorganisms-983175 “Genetic adaptation of Coxsackievirus B1 during persistent infection in pancreatic cells.” Corresponding author: Heikki Hyoty.

The authors investigate persistent CVB infection in pancreatic cells. They find several mutations consistent among laboratory and clinical strains corresponding to persistent infection.

Comments:

  1. 1: Why do the top two graphs begin at 2 – 4 weeks post infection instead of day 1? Is this a typo? If not how do genome copies during persistency compare to the initial phase of infection?
  2. 2: This data should be quantified. An average of ~10% positive cells is given, but what is the variation since only one representative field is shown?
  3. 3: The authors show that CVB1 ATCC doesn’t fit the trend. However, I don’t see any plaques at all in the top row. Why are there no plaques at earlier times but they appear at later timepoints?

Reviewer 2 Report

Honkimaa et al. present an also clinically relevant experimental study analysing the viral persistency of Coxsackie B (CVB)-1 strains and its potential role of inducing type 1 diabetes. The study is well designed and includes several methodological approaches to support the authors findings, where changes in the capsid region were identified. The manuscript is well written, the results clearly presented and discussion includes important papers of the specific topic.  

Specific comments:

-The authors describe in the result section (Figure 1) the viral growth in the cell and describe the number of viral copies in the supernatant. Please specifiy the multiplicity of infection (infection dose), show the cytotoxicity of the host cells. Moreover, the authors need to differentiate between viral copies within the cells and the supernatant. This should be more clearly displayed. Persistency occurs in the cells and not the supernatant.

-Is the replication of the strains in both cell lines different at different time points? The authors show only the total copy numbers from supernatants.

-Please explain, why less cells are infected at a later time point the 1.1B4 cells (Figure 2)

-What time points are shown in figure 2?

-In figure 3 some strains show discrepant effect on plaque formation. Please explain.

-Have the authors used the mutated strains to analyse replication capacity or other effects as cytotoxicity in the host cells?

Round 2

Reviewer 2 Report

The authors addressed the reviewer comments sufficiently, but the supplemental tables are missing.

Author Response

Dear Reviewer,

I apologise for my mistake not to submit the supplemetary material. It is included in the attachment.

Kind regards,

Anni Honkimaa
